# Mapping the Proteome of the Synaptic Cleft through Proximity Labeling Reveals New Cleft Proteins

**DOI:** 10.3390/proteomes6040048

**Published:** 2018-11-28

**Authors:** Tony Cijsouw, Austin M. Ramsey, TuKiet T. Lam, Beatrice E. Carbone, Thomas A. Blanpied, Thomas Biederer

**Affiliations:** 1Department of Neuroscience, Tufts University School of Medicine, Boston, MA 02111, USA; Bea.Carbone@tufts.edu; 2Department of Physiology and Program in Neuroscience, University of Maryland School of Medicine, Baltimore, MD 21201, USA; austin.ramsey@umaryland.edu (A.M.R.); tblanpied@som.umaryland.edu (T.A.B.); 3Yale/NIDA Neuroproteomics Center, New Haven, CT 06511, USA; tukiet.lam@yale.edu; 4W.M. Keck Biotechnology Resource Laboratory, Yale University School of Medicine, 300 George Street, New Haven, CT 06511, USA; 5Department of Molecular Biophysics and Biochemistry, Yale University School of Medicine, New Haven, CT 06520, USA

**Keywords:** synapse, synaptic cleft, trans-synaptic adhesion, proximity labeling, SynCAM, Cadm, Receptor-type tyrosine-protein phosphatase zeta, R-PTP-zeta, Ptprz1

## Abstract

Synapses are specialized neuronal cell-cell contacts that underlie network communication in the mammalian brain. Across neuronal populations and circuits, a diverse set of synapses is utilized, and they differ in their molecular composition to enable heterogenous connectivity patterns and functions. In addition to pre- and post-synaptic specializations, the synaptic cleft is now understood to be an integral compartment of synapses that contributes to their structural and functional organization. Aiming to map the cleft proteome, this study applied a peroxidase-mediated proximity labeling approach and used the excitatory synaptic cell adhesion protein SynCAM 1 fused to horseradish peroxidase (HRP) as a reporter in cultured cortical neurons. This reporter marked excitatory synapses as measured by confocal microcopy and was targeted to the edge zone of the synaptic cleft as determined using 3D dSTORM super-resolution imaging. Proximity labeling with a membrane-impermeant biotin-phenol compound restricted labeling to the cell surface, and Label-Free Quantitation (LFQ) mass spectrometry combined with ratiometric HRP tagging of membrane vs. synaptic surface proteins was used to identify the proteomic content of excitatory clefts. Novel cleft candidates were identified, and Receptor-type tyrosine-protein phosphatase zeta was selected and successfully validated. This study supports the robust applicability of peroxidase-mediated proximity labeling for synaptic cleft proteomics and its potential for understanding synapse heterogeneity in health and changes in diseases such as psychiatric disorders and addiction.

## 1. Introduction

Synapses are the cellular units for information transfer in the central nervous system. The mammalian brain is comprised of functionally diverse synapse types connecting different neuronal populations into networks that enable complex behavior and responses to external and internal cues. Understanding this diversity will be fundamental to defining brain connectivity [1,2,3]. Synapse function is instructed by the diverse proteomic composition of these specialized neuronal contact sites. These molecular components guide synaptogenesis, maturation, and differentiation in development and in adulthood [4,5]. Function and composition are specific to neuron identity, local connectivity, and regional specificity and contribute to a diversified and specialized synapse proteome within the brain [3]. The molecular diversity of synapses is further controlled by processes such as neuronal activity and plasticity changes, as well as by secreted factors [2,6,7]. Different states of the diseased brain, e.g., neuro-degeneration or addiction, may further alter synapse composition and function [8,9,10,11].

Synapses are comprised of a presynaptic terminal and a postsynaptic compartment, each containing protein complexes unique to their function [12,13]. Importantly, the pre- and post-synapse are connected through the synaptic cleft, which is a protein-dense environment organized into molecularly distinct sub-domains [13,14]. Biochemical fractionation methods combined with mass spectrometry-based proteomics have been instrumental in determining the general composition of central nervous system synapses. These approaches have allowed for detailed studies of synaptosomes, which contain presynaptic terminals with postsynaptic sites remaining attached [15], presynaptic membranes [11], presynaptic vesicles [16,17], active zones [18,19], and postsynaptic specializations [20,21,22]. The combination of these methods with the use of genetic models to tag synaptic components has allowed for the specific isolation of excitatory synaptic compartments [15,23] or inhibitory complexes [24]. While these classical biochemical studies have mapped general synapse composition, the ability to parse out their heterogeneity has been limited, and the synaptic cleft as a separate compartment was intractable. 

Recent proteomic advances have employed biotin-tagging of endogenous proteins within a specific cellular compartment using targeting of exogenous peroxidases that can create biotin-phenoxyl radicals or biotin ligases (BioID) without the need for cellular fractionation [25,26,27]. The enzyme horseradish peroxidase (HRP) and engineered peroxidases APEX and APEX2, derivatives from pea ascorbate peroxidase, have been used to map the proteome of mitochondria [28] and excitatory and inhibitory cleft proteomes [29]. These peroxidase-based labeling approaches are currently limited to cultured cells or small organisms that have been made permeable [30,31]. BioID involves the use of a promiscuous biotin ligase to biotinylate proximal proteins and has been used in mouse brain to map the proteome of inhibitory postsynaptic proteins [32]. However, this method requires hours to days of exogenous biotin application and the reactive biotin species has a relatively long lifetime. Recently, TurboID was developed using directed evolution and this engineered BioID mutant has enabled faster proximity labeling. Despite this advance, the temporal resolution is in the order of tens of minutes and background biotinylation is observed due to utilization of endogenous biotin by TurboID [33]. This limits the temporal and presumably the spatial resolution in contrast to peroxidase-mediated proximity labeling, which has a temporal resolution of tens of seconds and high ~20 nm spatial resolution due to the short lifetime of the enzyme-generated biotin radical [25].

Peroxidase-mediated proximity labeling has for the first time allowed to specifically dissect the proteome of the synaptic cleft of either excitatory or inhibitory synapses [29]. Loh, Ting and colleagues designed excitatory and inhibitory-specific cleft reporter proteins by fusion of HRP with excitatory synapse-specific LRRTM1 and LRRTM2 and inhibitory synapse-specific Neuroligin 2A and Slitrk3, and the authors made use of a ratiometric labeling technique for non-membrane enclosed cellular compartments. Biotinylated proteins of cleft-targeted HRP were identified and quantified using mass spectrometry and ratiometrically compared with biotinylated proteins of membrane-targeted HRP to detect cleft-enriched synaptic proteins. This resulted in proteomic lists representative of excitatory and inhibitory synapses, with a higher specificity and deeper coverage of trans-synaptic proteins than previous proteomes obtained after biochemical fractionation. 

Among the prominently expressed synaptic cleft proteins are SynCAMs, a group of immunoglobulin molecules that engage in homo- and heterophilic interactions [34,35]. They are specific for excitatory synapses [29,34] and instruct the formation and guide maturation of these synapses [4,34,36,37]. The current study describes the use of peroxidase-mediated proximity labeling to map the proteome of excitatory synapses using SynCAM 1 as a reporter protein. We describe the use of SynCAM 1-HRP-mediated labeling and in silico filtering steps to select synaptic proteins over generic plasma membrane proteins and intracellular contaminants. This resulted in a list of proteins that were each enriched in multiple biological replicate experiments and spanned functional categories expected to be present in the synaptic cleft. Identified proteins included synaptic proteins that were reported earlier using classical biochemical and peroxidase-mediated proximity labeling approaches. In addition, several proteins on our list are novel synaptic cleft candidates that may add to the parts list of the excitatory synaptic cleft. As part of our validation approach, we show that the trans-membrane protein Receptor-type tyrosine-protein phosphatase zeta, or R-PTP-zeta, identified here as a synaptic cleft candidate is prominently expressed across forebrain regions of the mouse brain and localized at excitatory synaptic sites. This study corroborates proximity labeling as an approach to map the cleft proteome and underlines its applicability to analyze synapse diversity.

## 2. Materials and Methods 

### 2.1 Animals

Pregnant Sprague Dawley timed-pregnancy rats were obtained from Charles River Laboratories (Wilmington, MA, USA). C57BL/6J background wild-type mice were obtained from Charles River Laboratories and maintained in the colony. All procedures were approved by the Institutional Animal Care and Use Committee (B2016-154) and in compliance with National Institutes of Health guidelines.

### 2.2 Neuronal Cell Culture

Dissociated cortical neuron cultures from embryonic day 18 (E18) rats were prepared as described previously [38]. In brief, pregnant rats were sacrificed and E18 embryos extracted. Embryo’s brains were dissected, and cortices were isolated. Cortices were incubated in 0.05% trypsin (Invitrogen 25300054, Carlsbad, CA, USA) at 37 °C for 20 min, triturated to single cell suspension, and plated on poly-l-lysine (Sigma-Aldrich P1274, St. Louis, MO, USA)-coated surfaces (i.e., 12-mm glass coverslips or 10-cm culture dishes). Cytosine arabinoside (Sigma-Aldrich C1768) was added at a final concentration of 2 μM for 2–4 days-in-vitro (div). For mass-spectrometry experiments, per condition 3 × 10^6^ cells/dish were plated on six times 10-cm cell culture dishes coated with poly-l-lysine. For immunocytochemistry, 100,000 cells were plated on 12-mm coverslips coated with poly-l-lysine in a well of a 24-well cell culture plate.

For dSTORM experiments, neuronal cell cultures were prepared as described before [14].

### 2.3. HEK293T Cell Culture

Human embryonic kidney (HEK) 293T cells (ATCC, Manassas, VA, USA) were seeded at 20% confluence in T75 flasks and cultured in Dulbecco’s Modified Eagle Medium (DMEM) (Gibco via Fisher Scientific, Hampton, NH, USA, #11-965-118) supplemented with 10% fetal bovine serum, penicillin, and streptomycin at 37 °C under 5% CO_2_. Cells were passaged at 80–90% confluence by trypsinization and reseeded. For biotinylation of HEK293T cells, 60,000 cells were reseeded to 12-mm uncoated coverslips.

### 2.4. Plasmids

Plasmid pCAGGS-SynCAM 1-APEX2 was cloned using general cloning procedures. Briefly, using restriction site *NheI*-flanking primers, FLAG-APEX2 (generating *NheI*-FLAG-APEX2-*NheI*) was amplified from plasmid pcDNA3-APEX2-NES (a gift from Alice Ting; Addgene plasmid #49386) and ligated into a pCR-BluntII-Topo vector. The resulting vector was then restriction digested using *NheI* and the excised fragment *NheI*-FLAG-APEX2-*NheI* was cloned into *NheI* restriction digested pCAGGS-SynCAM 1-(363-*NheI*) [34] resulting in pCAGGS-SynCAM 1-APEX2 with APEX2, N-terminal flanked with a FLAG-tag, at amino acid (AA) position 363 of mouse SynCAM 1.

Plasmid pAAV-CaMKIIa-HRP-TM (Membrane-TM) was cloned using general cloning procedures from pCAG-HRP-TM (a gift from Alice Ting; Addgene plasmid #44441). Briefly, pCAG-HRP-TM was restriction digested using *BamHI* and *HindIII*, and the excised HA-HRP-Myc-TM fragment was cloned into *BamHI* and *HindIII* restriction digested pAAV-CaMKIIa-EGFP (a gift from Bryan Roth; Addgene plasmid #50469), which removed EGFP but kept the CaMKIIa promoter. 

Initially, a SynCAM 1-HRP version was cloned that had insufficient biotinylation activity in neurons (not shown), presumably due to a lack of flexible linkers adjacent the HRP. This initial plasmid pAAV-CaMKIIa-SynCAM 1-HRP was assembled using the NEBuilder High-Fidelity DNA Assembly Cloning kit (New England BioLabs, Ipswich, MA, USA, E5520S) according to the manufacturer’s instructions and general cloning procedures. In brief, the fragments for the Gibson/Seamless cloning were: pAAV-CaMKIIa-EGFP (a gift from Bryan Roth; Addgene plasmid #50469) restriction digested using *BamHI* and *EcoRV* to remove EGFP, which served as vector backbone; 5′-fragment of SynCAM 1 containing amino acids 1–362 (of mouse SynCAM 1) amplified from pCR-BluntII-TOPO SynCAM 1(363-*NheI*) (see below); HRP (omitting start codon, HA-tag, and TM) amplified from pCAG-HRP-TM with 3′ of HRP a FLAG-tag (DYKDDDDKA) was introduced using additional sequences in the primers; and 3′ fragment of SynCAM 1 containing amino acids 363–445 (of mouse SynCAM 1) amplified from pCAGGS-SynCAM 1-(363-*NheI*) [34].

Then, a new SynCAM 1-HRP version was cloned that contained flanking linker sequences that sterically separated HRP-FLAG from SynCAM 1. This linker-containing plasmid pAAV-CaMKIIa-SynCAM 1-HRP was assembled using the NEBuilder High-Fidelity DNA Assembly Cloning kit (New England BioLabs, Ipswich, MA, USA, E5520S) according to the manufacturer’s instructions and general cloning procedures. In brief, the fragments for the Gibson/Seamless cloning were: pAAV-CaMKIIa-EGFP (a gift from Bryan Roth; Addgene plasmid #50469) restriction digested using *BamHI* and *HindIII* to remove EGFP, which served as vector backbone; 5′-fragment of SynCAM 1 containing amino acids 1–362 amplified from pCR-BluntII-TOPO SynCAM 1(363-*NheI*) (see below); HRP-FLAG amplified from the initial pAAV- CaMKIIa-SynCAM 1-HRP plasmid with 5′ of HRP three repeats of a GGGGS-linker added and 3′ of the FLAG-tag three repeats of a GGGS-linker added using additional sequences in the primers; and 3′ fragment of SynCAM 1 containing amino acids 363–445 (of mouse SynCAM 1) amplified from the initial pAAV-CaMKIIa-SynCAM 1-HRP plasmid. The resulting plasmid pAAV-CaMKIIa-SynCAM 1-HRP, containing HRP-FLAG at AA position 363 of mouse SynCAM 1 and flanked with linkers, was used as SynCAM 1-HRP in all neuronal studies presented here.

The 5′-fragment of SynCAM 1 used in the Gibson assemblies above originated from a template where a *BamHI* restriction site at base pair position 25 (from start of coding sequence) in SynCAM 1 was mutated (synonymous) from GGATCC to GGTTCC using site-directed mutagenesis: pCR-BluntII-TOPO SynCAM 1(363-*NheI*).

Plasmids pCAGGS-SynCAM 1-APEX2 (Plasmid ID 119727), pAAV-CaMKIIa-SynCAM 1-HRP (Plasmid ID 119728), and pAAV-CaMKIIa-HRP-TM (Plasmid ID 119729) are available at Addgene.org.

### 2.5. Adeno-Associated Virus Production, Purification, and Titration

AAV was produced using the triple-transfection, helper-free method, with a modified version of a published protocol [39,40]. Briefly, AAV-293 cells (gift from Ralph DiLeone, Yale University, New Haven, CT, USA), a HEK-293-based cell line optimized for the packaging of AAV virions, were cultured in five 150-mm diameter cell culture dishes and transfected with pAAV-Reporter (i.e., HRP-fusion proteins), pHelper (gift from Ralph DiLeone, Yale University, New Haven, CT, USA), and pAAV-DJ-Rep-Cap (gift from Pascal Kaeser, Harvard University, Cambridge, MA, USA) plasmids using the acidified polyethylenimine (PEI; Polysciences, Inc., Warrington, PA, USA, 23966-2) method [41]. Cells were collected, pelleted, and resuspended in freezing buffer (0.15 M NaCl, 50 mM Tris, pH 8.0) 48–72 h after transfection. After four freeze-thaw cycles using liquid nitrogen and a 42 °C water bath, benzonase was added (Sigma-Aldrich, E1014; 50 U/mL, final) and incubated at 37 °C for 30 min. The lysate was spun at 3200× *g* for 30 min at 4 °C, and supernatant was added to an Optiseal centrifuge tube (Beckman Coulter, 361625, Brea, CA, USA) containing a 15%, 25%, 40%, and 60% iodixanol (Optiprep, 60%; Sigma-Aldrich, D1556) step gradient. The lysate on the step gradient was spun at 184,000× *g* (RCF average) for 3 h and 20 min at 10 °C (50,000 rpm, Beckman Optima LE-80K, Type 70 Ti Beckman rotor, Beckman Coulter) and the 40% fraction was collected. Iodixanol buffer solution was exchanged and AAV concentrated with 1× PBS containing 1 mM MgCl_2_ and 2.5 mM KCl (PBS-MK) using Amicon Ultra centrifugal filters (100,000 NMWL;, UFC910024, Merck Millipore, Burlington, MA, USA). The purified virus was stored at −80 °C. 

To titrate AAV, various quantities of purified virus were added to cultured neurons at 14–17 div. At 21–24 div, neurons were labeled and imaged as described below. Virus titer amount was selected based on the criteria that biotinylation was visible at distinct puncta for SynCAM 1-HRP at sites of Homer or was diffusely along the membrane for Membrane-HRP and overall transduction efficiency was >50%. At these expression levels, the FLAG and HA-antibodies to detect SynCAM 1-HRP or Membrane-HRP, respectively, were generally not sensitive enough to detect the reporters in immunocytochemistry and biotinylation served as marker of these reporters.

### 2.6. Transfection

For dSTORM imaging of SynCAM 1-HRP, cultured neurons on coverslips were transfected at 18 div using lipofectamine 2000 (1 µg/µL DNA) (Thermo Fisher Scientific, Waltham, MA, USA) and 1 µg/coverslip total pAAV SynCAM 1-HRP DNA in 50 µL opti-mem (Thermo Fisher Scientific) per coverslip. DNA was first added to half the total volume of opti-mem, subsequently pipetted into the other half the total volume of opti-mem containing lipofectamine and allowed to incubate for 5–20 min. 50 µL of the mixture was then pipetted drop-wise into each well containing a coverslip.

For HEK293T cell biotinylation and immunocytochemistry, APEX2 or HRP-fusion constructs were introduced by the acidified polyethylenimine (PEI; 23966-2, Polysciences, Inc., Warrington, PA, USA) method [41] one day after seeding.

### 2.7. Peroxidase-Mediated Biotinylation

For each specific neuron labeling experiment (i.e., neuronal cell biotinylation and immunocytochemistry, neuronal cell biotinylation and Western blot staining, and neuronal cell biotinylation and mass spectrometry), at indicated days cells were labeled live with 100 μM membrane-impermeant biotin-AEEA-phenol (Iris Biotech, Marktredwitz, Germany, LS-3490.0100) and 1 mM H_2_O_2_ (Sigma-Aldrich, 95321) in Tyrode’s buffer (145 mM NaCl, 1.25 mM CaCl_2_, 3 mM KCl, 1.25 mM MgCl_2_, 0.5 mM NaH_2_PO_4_, 10 mM glucose, 10 mM HEPES (pH 7.4)) for 1 min at room temperature. After 1 min, the biotinylation reaction was quenched by washing the cells three times with Tyrode’s buffer containing 10 mM sodium azide (#BDH7465-2, VWR, Radnor, PA, USA), 10 mM sodium ascorbate (Sigma-Aldrich, #PHR1279), and 2.5 mM Trolox (Acros Organics via VWR, #200008-026) as described [29]. Neurons for immunocytochemistry were fixed with 4% paraformaldehyde in “fixation buffer” (60 mM PIPES, 25 mM HEPES, 10 mM EGTA, 2 mM MgCl_2_, 0.12 M sucrose [pH 7.4]) at room temperature for 10 min. Neurons for Western blot or mass spectrometry were immediately harvested, frozen in liquid nitrogen, and stored at −80 °C awaiting further processing.

For HEK293T cell biotinylation and immunocytochemistry, 1–2 days after transfection cells were labeled live with by adding a final concentration of 100 μM membrane-impermeant biotin-AEEA-phenol and 1 mM H_2_O_2_ to the cell culture for 1 min or by adding 500 μM membrane-permeant biotin-phenol (Iris Biotech, Marktredwitz, Germany, #LS-3500.0250 ) to the cell culture medium for 30 min at 37 °C and afterwards adding a final concentration of 1 mM H_2_O_2_ to the cell culture medium. Then, HEK293T cells were washed three times with PBS containing 10 mM sodium azide, 10 mM sodium ascorbate, and 2.5 mM Trolox to quench the biotinylation reaction, followed by fixation with 4% paraformaldehyde/4% sucrose in PBS for 10 min at room temperature. 

### 2.8 Sample Preparation for Mass Spectrometry

Cell pellets were thawed on ice and each pellet was resuspended in 350 μL lysis buffer (1% SDS in 50 mM Tris-HCl (pH 8.0), including 10 mM sodium azide, 10 mM sodium ascorbate, and 2.5 mM Trolox, and the protease inhibitors at a final concentration of 1 mM PMSF, 2 μg/mL leupeptin, 1 μg/mL pepstatin, and 1 μg/mL aprotonin). Lysates were boiled at 95 °C for 5 min to dissociate the postsynaptic density (PSD) and diluted with 1400 μL 1.25× RIPA lysis buffer (50 mM Tris, 187.5 mM NaCl, 0.625% sodium deoxycholate, 1.25% Triton X-100) to a final 1× RIPA lysis buffer (50 mM Tris-HCl [pH 8.0], 150 mM NaCl, 0.2% SDS, 0.5% sodium deoxycholate, 1% Triton X-100). Lysates were cleared by centrifugation at 16,000× *g* for 10 min at 4 °C. Streptavidin-coated magnetic beads (Dynabeads M-270, Thermo Fisher Scientific, 65305) were equilibrated to room temperature and 50 μL resuspended bead slurry was washed twice with 1× RIPA buffer 50 mM Tris-HCl [pH 8.0], 150 mM NaCl, 0.2% SDS, 0.5% sodium deoxycholate, 1% Triton X-100). Beads were incubated with the lysate overnight at 4 °C with gentle rocking agitation to bind biotinylated proteins. Beads were then washed four times with 1 mL ice-cold 1× RIPA buffer followed by two washes with 1 mL 100 mM ammonium bicarbonate. In the second wash a final concentration of 0.01% RapiGest SF (#186001861, Waters Corporation, Milford, MA, USA) was added to avoid beads adhering to the tube.

For tryptic digestion of the bound biotinylated proteins, beads were incubated with 10 μL of 10 ng/μL trypsin (Sigma-Aldrich, 11418475001) in 100 mM ammonium bicarbonate containing 0.1% RapiGest SF overnight at 37 °C. For the first 15 min, samples were vortexed for 15 s every 2–3 min. After overnight digestion, another 10 μL of 10 ng/μL trypsin in 100 mM ammonium bicarbonate containing 0.1% RapiGest SF was added and incubated for 4 h at 37 °C. Afterwards, 100% formic acid (Fisher Chemical, Fisher Scientific, Hampton, NH, USA, A117-50) was added to a final concentration of 5% (*v/v*). Samples were frozen in liquid nitrogen and stored at −80 °C before shipment on dry ice to a mass spectrometry facility.

### 2.9 Mass Spectrometry and Label-Free Quantification

#### 2.9.1. Mass Spectrometry Data Acquisition

The acidified samples were placed in an autosampler vial for analysis by reverse phase liquid chromatography-tandem mass spectrometry (RP)-LC-MS/MS. Reversed phase (RP)-LC-MS/MS was performed using nanoACQUITY UPLC system (Waters Corporation, Milford, MA, USA) connected to an Orbitrap Fusion Tribrid (Thermo Fisher Scientific, San Jose, CA, USA) mass spectrometer. After injection, samples were loaded into a trapping column (nanoACQUITY UPLC Symmetry C18 Trap column, 180 µm × 20 mm) at a flowrate of 5 µL/min and separated with a C18 column (nanoACQUITY column Peptide BEH C18, 75 µm × 250 mm). The compositions of mobile phases A and B were 0.1% formic acid in water and 0.1% formic acid in acetonitrile, respectively. Peptides were eluted with a gradient extending from 3% to 35% mobile phase B in 90 min at a flowrate of 300 nL/min and a column temperature of 37 °C. The data were acquired with the mass spectrometer operating in a top speed data-dependent mode. The full scan was performed in the range of 300–1500 *m/z* at an Orbitrap resolution of 120,000 at 200 *m/z* and automatic gain control (AGC) target value of 4 × 10^5^. Full scan was followed by MS^2^ event of the most intense ions above an intensity threshold of 5 × 10^4^. The ions were iteratively isolated with a 1.6 Th window, injected with a maximum injection time of 110 msec, AGC target of 1 × 10^5^, and fragmented with higher-energy collisional dissociation (HCD). 

#### 2.9.2. Data Processing for Identification

Raw data from the Orbitrap Fusion mass spectrometer (Thermo Fisher Scientific, Waltham, MA, USA) were processed using Proteome Discoverer software (version 2.1, Thermo Fisher Scientific). MS^2^ spectra were searched using Mascot (Matrix Science, Boston, MA, USA) which was set up to search against the SwissProt rat database. The search criteria included 10 ppm precursor mass tolerance, 0.02 Da fragment mass tolerance, trypsin as proteolytic enzyme, and maximum missed cleavage sites of two. Potential dynamic modifications assigned were oxidation of methionine, de-amidation of asparagine, acetylation of N-terminus, and phosphorylation of serine, threonine and tyrosine.

Scaffold (version Scaffold_4.8.4, Proteome Software Inc., Portland, OR, USA) was used to validate MS/MS based peptide and protein identifications. Peptide identifications were accepted if they could be established at greater than 95.0% probability by the Peptide Prophet algorithm [42] with Scaffold delta-mass correction. Protein identifications were accepted if they could be established at greater than 99% probability and contained at least two identified peptides. Protein probabilities were assigned by the Protein Prophet algorithm [43]. Proteins that contained similar peptides and could not be differentiated based on MS/MS analysis alone were grouped to satisfy the principles of parsimony. Proteins sharing significant peptide evidence were grouped into clusters. 

The mass spectrometry proteomics data have been deposited to the ProteomeXchange Consortium via the PRIDE partner repository [44] with the dataset identifier PXD011312. 

#### 2.9.3 Post Hoc In-Silico Filtering/Data Analysis

Data of all four biological replicate experiments were loaded together into Scaffold 4 (version 4.8.4) and GO annotations were added (Appendix A) (NCBI, downloaded 20 December 2017). Per each biological replicate experiment, five conditions were included that each represented one sample (Figure 3B): non-transduced rat cortical neurons treated with biotin-AEEA-phenol and H_2_O_2_ (condition 1, c1); neurons transduced with Membrane-HRP rAAV and treated with biotin-AEEA-phenol omitting H_2_O_2_ (condition 2, c2) or with H_2_O_2_ (condition 3, c3); neurons transduced with SynCAM 1-HRP rAAV and treated with biotin-AEEA-phenol omitting H_2_O_2_ (condition 4, c4) or with H_2_O_2_ (condition 5, c5). Quantitative values, i.e., iBAQ (intensity-Based Absolute Quantitation) values [45] (no normalization), of detected proteins were calculated and exported to Microsoft Excel in which all further analysis was proceeded. Proteins identified within sub-clusters, keratin (a known impurity), and proteins with a probability <95% were excluded and protein iBAQ values were normalized to summed iBAQ values within one sample representing the molar abundance [46] or relative iBAQ (riBAQ) of an identified protein within a sample. Filtering was based upon previous protocol [29] for Filter 1 and 2 and modified as follows.

##### Filter 1

For each detected protein, the log_2_ of c5/c1 riBAQ value within a biological replicate was calculated, which measured the extent of biotinylation by a reporter protein. If a protein was not detected in c1 and hence a ratio could not be calculated the protein was regarded to be detected with high specificity and received the label “Specific”. If a protein was not detected in c5, it received the label “N.A.”. For Filter 1, in each biological replicate a c5/c1 riBAQ ratio cut-off value was determined above which a protein was retained using Receiver Operating Characteristic (ROC) analysis. First, proteins were labeled according to the following four groups [29]: (1) true positive or known synaptic proteins, TP1; (2) false positives or known intracellular proteins, FP1; (3) known surface proteins, FP2; (4) all other proteins. Proteins that existed in multiple groups were re-sorted to group 4. Endogenously biotinylated proteins and proteins with a cell surface GO-term were excluded from group 2. The c5/c1 riBAQ ratios of group 1 and group 2 were plotted in histograms (Figure 4B). An ROC curve analysis was then performed (Figure 4C) [47]. The remaining proteins were ranked in descending order according to the c5/c1 riBAQ ratio and then the riBAQ value. Proteins with “Specific” values ware placed on top and then ranked according to riBAQ values, and “N.A.” proteins were excluded. In the ranked list, the True Positive Rate (TPR) for each protein was calculated as the summed number of group 1 proteins up till (and including) that protein divided by the total number of proteins in group 1. The False Positive Rate (FPR1) for each protein was calculated as the summed number of group 2 proteins up till that protein divided by the total number of proteins in group 2. For each ranked protein, TPR-FPR1 was calculated and plotted against its rank (Figure 4C). At maximum TPR-FPR, the associated ranked protein was found and its log_2_ of c5/c1 riBAQ value determined.

##### Filter 2

For each detected protein, the log_2_ c5/c3 riBAQ ratio within a biological replicate was calculated, which measured the extent of biotinylation by excitatory cleft-localized SynCAM 1-HRP over dendritic membrane-localized Membrane-HRP. 

If a protein was not detected in c3 and hence a ratio could not be calculated the protein was regarded to be detected with high specificity and received the label “Specific”. If a protein was not detected in c5, it received the label “N.A.”. For Filter 2, in each biological replicate a c5/c3 riBAQ ratio cut-off value was determined above which a protein was retained using Receiver Operating Characteristic (ROC) analysis. First, proteins were labeled according to the following four groups [29]: (1) true positive or known synaptic proteins, TP1; (2) false positives or known intracellular proteins, FP1; (3) known surface proteins, FP2; (4) all other proteins. Proteins that existed in multiple groups were re-sorted to group 4. Endogenously biotinylated proteins and proteins with a cell surface GO-term were excluded from group 2. The c5/c3 riBAQ ratios of group 1 and group 3 were plotted in histograms (Figure 4B). An ROC curve analysis was then performed (Figure 4C) [47]. The remaining proteins were ranked in descending order according to the c5/c3 riBAQ ratio and then the riBAQ value. Proteins with “Specific” values ware placed on top and then ranked according to riBAQ values, and “N.A.” proteins were excluded. In the ranked list, in descending order the True Positive Rate (TPR) for each protein was calculated as the summed number of group 1 proteins found from the top up till (and including) that protein divided by the total number of proteins in group 1. The False Positive Rate (FPR2) for each protein was calculated as the summed number of group 3 proteins found up till that protein divided by the total number of proteins in group 3. For each ranked protein, TPR-FPR2 was calculated and plotted against its rank (Figure 4C). At maximum TPR-FPR2, the associated ranked protein was found and its log_2_ of c5/c3 riBAQ value determined.

##### Filter 3

For each biological replicate, cut-off values for Filter 1 and Filter 2 were applied. Identified proteins in a biological replicate were retained when above the cut-off value or when labeled “Specific” for that filter. All identified proteins were then ranked in descending order to how many biological replicates they remained after filtering, how often they were labeled “Specific”, and the average c5/c3 riBAQ ratio of all biological replicates. A ROC curve analysis was then performed (Figure 4F). In the ranked list, in descending order the True Positive Rate (TPR) for each protein was calculated as the summed number of group 1 proteins found from the top up till (and including) that protein divided by the total number of proteins in group 1. The False Positive Rate (FPR2) for each protein was calculated as the summed number of group 3 proteins found up till that protein divided by the total number of proteins in group 3. For each ranked protein, TPR-FPR2 was calculated and plotted against its rank (Figure 4F). 

In the final selection step, proteins were removed that had only GO-terms associated with intracellular locations and proteins on the False Positive 1 (FP1)-list [29]. Serum albumin was removed from Appendix A as it is a likely carry-over from cell culture medium.

#### 2.9.4. Calculation of Depth of Coverage

Depth of coverage for the excitatory synaptic cleft proteome was calculated as in [29]. In total, 10 excitatory synaptic cleft candidate proteins identified here (Appendix A) were among the list of 62 literature identified excitatory synaptic cleft proteins (TP2, Appendix A).

### 2.10. Immunochemistry

For visualization of biotinylation in HEK293T cells, after the biotinylation reaction and fixation (see Peroxidase-mediated biotinylation), cells were washed three times with PBS and permeabilized with 0.1% Triton X-100 for 10 min at room temperature followed by blocking with 5% FBS in PBS 1 h at room temperature. Then, cells were incubated for 1 h at room temperature with Streptavidin-Alexa488 (Molecular Probes, S11223; 1:500) in 5% FBS in PBS, followed by three washes with PBS. SynCAM 1-APEX2 or APEX2-NES were detected by immunostaining using anti-FLAG M2 IgG1 antibodies raised in mouse (Sigma-Aldrich, F1804; 1:500) or anti-HA rabbit antibody (Cell Signaling Technologies, #3724; 1:500), resp., for 1 h at room temperature followed by three washes with PBS and secondary antibody staining with anti-IgG1 Alexa568-conjugated secondary antibodies (Invitrogen via Fisher, #A21124; 1:500) or anti-rabbit Alexa568-conjugated secondary antibodies (Invitrogen via Fisher, #A11036; 1:500), resp., for 1 h at room temperature followed by three washes with PBS. Coverslips were mounted onto glass slides (Aqua-Poly/Mount, Polysciences) and imaged with a Leica SP2 confocal microscope (Leica Camera Co., Wetzlar, Germany).

For biotinylation in neurons, after the biotinylation reaction and fixation (see Peroxidase-mediated biotinylation), non-permeabilized cells were blocked with 5% BSA (RMBIO, BSA-BAF) for 30 min at 4 °C followed by staining for biotin using NeutrAvidin protein, Dylight 488 (NA-488; Thermo Fisher Scientific, 22832; 1:250) for 10 min at 4 °C, immediately followed by one wash with 1 mM free biotin (Sigma-Aldrich, #B4501) in PBS and two washes in PBS. Staining with more concentrated NA-488 at 4 °C to reduce membrane protein mobility, followed with a biotin wash to block free biotin-binding sites in bound NA-488 was necessary to avoid that biotin-labeled surface proteins clustered in the membrane [48]. Cells were permeabilized with 0.1% Triton X-100 for 10 min at room temperature followed by blocking with 5% FBS in PBS 1 h at room temperature. From here on all permeabilization steps were with 0.1% Triton X-100 for 10 min, blocking was with 5% FBS in PBS 1 h at room temperature, and all primary and secondary stainings were done at room temperature for 1 h or overnight at 4 °C in 5% FBS in PBS and followed by three times 5-min washes with PBS at room temperature. Cells were stained with antibodies raised in rabbit against Homer (Synaptic Systems GmbH, Goettingen, Germany, 160 003; 1:500) and anti-rabbit Alexa568 secondary antibodies (1:500). Coverslips were mounted onto glass slides (Aqua-Poly/Mount, Polysciences) and imaged with a Leica SP8 confocal microscope (Leica Camera Co., Wetzlar, Germany). 

For visualization of R-PTP-zeta in neurons, dissociated cortical neuron cultures from rats were fixed with fixation buffer (60 mM PIPES, 25 mM HEPES, 10 mM EGTA, 2 mM MgCl_2_, 0.12 M sucrose [pH 7.4]) at 4 °C for 15 min and washed three times with PBS before blocking. Under non-permeabilizing conditions, R-PTP-zeta and SynCAM 1 were stained with anti-PTPζ IgM antibodies raised in mouse (Santa Cruz Biotechnology, Dallas, TX, USA, sc-33664; 1:500) and anti-SynCAM 1 antibodies raised in chicken (MBL Laboratories, Woods Hole, MA, USA), CM004-3; 1:500), respectively. Followed by staining with anti-IgM Alexa488 secondary antibodies (Life Technologies, Carlsbad, CA, USA, A21042; 1:500) and anti-chicken Alexa647 secondary antibodies (Life Technologies, A21449; 1:500). Cells were permeabilized followed by blocking. Cells were then either stained for Bassoon or Homer with anti-Bassoon mouse IgG2A antibodies (Enzo Life Sciences, Enzo Biochem., Inc., Farmingdale, NY, USA, ADI-VAM-PS003; 1:750) or anti-Homer rabbit (Synaptic Systems, 160 003; 1:500), respectively, and then anti-IgG2a Alexa568 (1:500) or anti-rabbit Alexa568 (1:500) secondary antibodies. Coverslips were mounted (Aqua-Poly/Mount, Polysciences) and imaged with a Leica SP8 confocal microscope (Leica Camera Co., Wetzlar, Germany).

For visualization of R-PTP-zeta in brain sections, adult mice were anaesthetized using ketamine/xylazine and perfused transcardially with 4% paraformaldehyde (PFA) in PBS. Brains were extracted and post-fixed overnight at 4 °C in 4% PFA in PBS. Brains were sectioned on a vibratome (Vibratome 1500, Leica Camera Co., Wetzlar, Germany) into 60 μm coronal sections. For detection of R-PTP-zeta, Homer, and Bassoon, individual sections were simultaneously permeabilized and blocked with 0.3% Triton X-100 and 3% horse serum for 1 h at room temperature, followed by a combined incubation with anti-PTPζ mouse IgM (Santa Cruz Biotechnology, sc-33664; 1:500), IgG2A antibodies raised against Bassoon in mouse (1:750) and anti-Homer antibodies raised in rabbit (1:500) (all in 0.3% Triton X-100 and 3% horse serum for 3 days at 4 °C). Followed by three washes with PBS and incubation with anti-IgM Alexa488 (1:500), anti-IgG2a Alexa647 (1:500) or anti-rabbit Alexa568 (1:500) antibodies in 0.3% Triton X-100 and 3% horse serum. Sections were mounted onto glass slides (Aqua-Poly/Mount, Polysciences, Inc., Warrington, PA, USA) and imaged with Keyence microscope (BZ-X700, Keyence, Itasca, IL, USA). Secondary antibodies were purchased from Life Technologies.

### 2.11. Characterization of Biotinylation by SynCAM 1-HRP Using Western Blot

Cell pellets were thawed on ice and each cell pellet was resuspended in 100 μL lysis buffer (1% SDS in 50 mM Tris-HCl (pH 8.0), including 10 mM sodium azide, 10 mM sodium ascorbate, and 2.5 mM Trolox, and the protease inhibitors at final concentration of 1 mM PMSF, 2 μg/mL leupeptin, 1 μg/mL pepstatin, and 1 μg/mL aprotonin). Lysates were boiled at 95 °C for 5 min to dissociate the postsynaptic density (PSD) and diluted with 400 μL 1.25× RIPA lysis buffer (50 mM Tris, 187.5 mM NaCl, 0.625% sodium deoxycholate, 1.25% Triton X-100) to a final 1× RIPA lysis buffer (50 mM Tris-HCl [pH 8.0], 150 mM NaCl, 0.2% SDS, 0.5% sodium deoxycholate, 1% Triton X-100). Lysates were cleared by centrifugation at 16,000× *g* for 10 min at 4 °C. Cleared lysates were then boiled in sample buffer (containing final 50 mM DTT and 2% SDS in 60 mM Tris-HCl, pH 6.8) for 3 min and separated on a 10% polyacrylamide gel. Proteins were then blotted onto nitrocellulose blotting membrane and stained with Ponceau S for equal loading confirmation. Membrane was blocked in 5% BSA in TBST and incubated with NeutrAvidin-DyLight488 (Thermo Fisher Scientific, #22832; 1:500) in 5% BSA in TBST for 1 h at room temperature, followed by three washes with TBST and imaged (FluorChem M, ProteinSimple, San Jose, CA, USA).

### 2.12. Staining for dSTORM, and dSTORM Imaging and Analysis of SynCAM 1-HRP

For dSTORM experiments, neurons were fixed in 4% PFA with sucrose for 15 min at 21 div then washed for 3 times for 5 min each in PBS containing glycine (PBS/Gly). Samples were blocked in donkey serum (10%) for 20 min. Surface labeling was done by inverting coverslips on 50 µL droplets containing the anti-FLAG M2 antibody (Mouse IgG1) at 1:500 on parafilm. Coverslips were washed two times in PBS/Gly. Permeabilization was achieved with 0.1% TX-100 for 20 min. This was exchanged with PBS/Gly containing 10% donkey serum and 0.1% TX-100 and incubated for an additional 20 min. Coverslips were then transferred and inverted onto 50 µL droplets on parafilm containing anti-Homer1 primary antibody (Synaptic Systems; 1:500) at 4 °C overnight (<15 h). Primary contains 0.03% TX-100 and 5% donkey serum. Coverslips were washed 3 times for 5 min each in PBS/Gly the next day. Coverslips were then transferred and inverted onto parafilm with 50 µL droplets containing secondary antibody for 1 h at room temperature in the dark (Donkey anti-Rabbit CF555, 1:500; Donkey α-Mouse Alexa-647, 1:200). Coverslips were then washed 3 times for 5 min each in PBS/Gly and were then imaged according to [14].

Images were denoised using Thunderstorm’s Wavelet (basis spline) filter ([49]; http://imagej.net/Fiji). Approximate localization of molecules was done using the local maximum approach using the Thunderstorm plug-in in Fiji/ImageJ (dev-2016-01-01-b1, github.com/zitmen/thunderstorm/). Sub-pixel localization of stochastic blinking was achieved by fitting peaks with an integrated Gaussian function using the Thunderstorm plug-in. For visualization, pixels were magnified 10× and detected localizations were binned into pixels. Poorly fit localizations were filtered out using the Thunderstorm software. Drift correction was performed in Thunderstorm using the cross-correlation method. The visualization method used is the Average Shifted Histograms method in Thunderstorm.

All synaptic data was analyzed by identifying synaptic clusters containing both Homer1 and SynCAM 1-HRP, in order to identify synapses from transfected cells. For line scan analyses, synaptic pairs were selected, then a line (20 pixels wide) was drawn perpendicular to the synaptic axis, as to pass through both Homer1 and SynCAM 1-HRP. SynCAM 1-HRP puncta were counted by eye in ImageJ. For the inter-cluster distance analysis, center was determined by eye and a line was drawn between the two clusters and the distance was estimated by rounding to the nearest pixel.

## 3. Results

### 3.1. Peroxidase-Mediated Proximity Labeling Using a SynCAM 1-Horseradish Peroxidase (HRP)-Fusion Protein 

Peroxidase-mediated proximity labeling allows for the biotinylation of endogenous proteins proximal to a recombinant peroxidase-reporter fusion protein that is exogenously expressed at a cellular compartment. Peroxidases that have been used for proximity labeling are HRP and APEX/APEX2, which are metalloenzymes that catalyze the oxidation of organic substrates by hydrogen peroxide (H_2_O_2_). These peroxidases can oxidize in presence of H_2_O_2_ biotin-phenol compounds to generate short-lived biotin-phenoxyl radicals. These radicals can form covalent bonds with tyrosine and other electron-rich amino acids at proximal proteins [25,50]. APEX2, a more active variant of the engineered peroxidase APEX [51], is suitable for intracellular and extracellular applications, while HRP is only active in the oxidizing environments of the secretory pathway and the cell surface due to its structurally essential disulfide-bonds. 

To map the excitatory synaptic cleft proteome, a fusion protein needs to target a peroxidase to this specific compartment. We here used SynCAM 1 as it is a synaptic cell adhesion protein that localizes exclusively to excitatory synapses during development and in mature synapses [25,34]. HRP or APEX2 was inserted at amino acid position 363 of SynCAM 1, located between the last immunoglobulin (Ig) domain and the trans-membrane (TM) domain. This placed HRP at the base of the extracellular region of SynCAM 1, creating a SynCAM 1-HRP fusion protein (Figure 1A) or SynCAM 1-APEX2 fusion protein. Recombinant proteins or tags inserted at this position do not alter the synaptic localization of SynCAM 1 [52]. SynCAM 1 is a single-pass type 1 membrane protein that after biogenesis requires trafficking through the secretory pathway to reach the cell surface. To restrict biotin labeling mediated by peroxidase-fusion proteins that traffic through the secretory pathway to the cell surface, including the synaptic surface, a membrane-impermeant biotin-phenol compound containing a polar linker is required [29]. We therefore selected the commercially available compound biotin-AEEA-phenol, which is membrane-impermeant. 

To verify that it allows for biotinylation only at the cell surface, biotin-AEEA-phenol was added to HEK293T cells that expressed a membrane-bound SynCAM 1-APEX2 proximity reporter. In presence, but not in absence, of H_2_O_2_, exogenous biotin-AEEA-phenol induced biotinylation at the surface of HEK293T cells expressing SynCAM 1-APEX2 (Figure 1B). As control whether this biotin compound is membrane impermeable, biotin-AEEA-phenol was added to HEK293T cells expressing soluble APEX2 fused to a nuclear export sequence (APEX2-NES) [51]. This did not result in detectable intracellular biotinylation (Figure 1C). As expected from a previous study [28], membrane-permeable biotin-phenol allowed for intracellular biotinylation in HEK293T cells expressing APEX2-NES (not shown). Biotin-AEEA-phenol hence restricts proximity labeling to the cell surface. To efficiently use the proximity labeling approach in cultured neurons, HRP fusion proteins were used in all proximity labeling experiments in neurons as HRP is more active than APEX2 in the extracellular space [29].

### 3.2. Subsynaptic Distribution of the SynCAM 1-HRP Reporter 

Previously, the synaptic expression of endogenous SynCAM 1 has been analyzed using STED and 3D dSTORM super-resolution imaging in cultured neurons and by immuno-EM in brain sections. These approaches determined that SynCAM 1 localizes to the synaptic cleft of excitatory, asymmetric synapses and is predominantly present in the postsynaptic membrane, where it was detected around the edge of the postsynaptic density [14]. To analyze the localization of the SynCAM 1-HRP reporter, a FLAG-tag was inserted C-terminal of the HRP. Cultured rat hippocampal neurons were transfected with this construct and analyzed by two-color 3D dSTORM super-resolution imaging. Immunostaining for FLAG to detect SynCAM 1-HRP and for the postsynaptic excitatory scaffolding protein Homer followed by 3D dSTORM showed that super-resolved SynCAM 1-HRP localized adjacent to clusters of Homer (Figure 2A and Appendix A). Distribution of SynCAM 1-HRP ensembles were within ~100 nm of the edge of the PSD as outlined by super-resolved Homer clusters (Figure 2B and Appendix A), in agreement with the sub-synaptic distribution of endogenous SynCAM 1 in cultured rat hippocampal neurons [14]. This demonstrated that the SynCAM 1-based reporter targets HRP to the synaptic cleft and is enriched at the cleft border zone.

### 3.3. SynCAM 1-HRP Validation for Synaptic Proximity Labeling

Synaptic clefts are open cellular compartments that are not enclosed by membranes. This requires to further improve the specificity of protein identification in the synaptic cleft. A ratiometric peroxidase-mediated labeling approach was previously applied to synaptic cleft proteins [29] and mitochondrial inter-membrane space proteins [55]. We aimed to implement this ratiometric approach and compared the proteins biotinylated by the SynCAM 1-HRP cleft reporter (Figure 2C) versus a broadly surface-expressed HRP fused extracellularly to a trans-membrane domain (Membrane-HRP; Figure 2E) [28]. To verify proper generation of biotinylation reaction products by these two reporters in neurons, the reporters were transduced into cultures of dissociated rat cortical neurons using recombinant AAV (rAAV) at 14 days-in-vitro (div). rAAV allowed the large-scale transduction of cultured neurons, and virus particles were titrated to balance high transduction efficiency with moderate protein overexpression. Neurons were subjected to brief peroxidase-mediated proximity labeling at 21 div upon addition of H_2_O_2_ and biotin-AEEA-phenol. Staining of biotinylation products using NeutrAvidin-DyLight488 visualized biotin-protein conjugates along dendrites at excitatory synaptic locations positive for Homer in SynCAM 1-HRP expressing neurons (Figure 2D), with some extra-synaptic labeling. In contrast, Membrane-HRP transduced into cultured rat cortical neurons that underwent proximity labeling showed biotin labeling along dendrites that was not enriched at synaptic locations (Figure 2F). These results supported that the SynCAM 1-HRP and the Membrane-HRP reporters can be used to obtain proximity-labeled protein samples from excitatory synaptic clefts and the neuronal cell surface, respectively, for comparative ratiometric analysis in silico.

### 3.4. Robust Identification of Synaptic Cleft Candidate Proteins Using Proximity Labeling 

To map the proteome of excitatory synaptic clefts, large-scale rat cortical cultures were prepared and treated following the work flow shown in Figure 3A. rAAV encoding SynCAM 1-HRP or Membrane-HRP was added to separate neuronal cultures at 14 div. Cultures underwent at 21 div peroxidase-mediated labeling upon addition of H_2_O_2_ and biotin-AEEA-phenol for 60 s. Biotinylated proteins were purified and peptides were generated by on-bead digestion with trypsin. Peptides were analyzed by Label-Free Quantitation (LFQ) proteomics, which compares peptide intensities across samples [56,57,58]. Per each biological replicate experiment, five conditions were included that each represented one sample (Figure 3B): non-transduced rat cortical neurons treated with biotin-AEEA-phenol and H_2_O_2_ (condition 1); neurons transduced with Membrane-HRP rAAV and treated with biotin-AEEA-phenol omitting H_2_O_2_ (condition 2) or with H_2_O_2_ (condition 3); neurons transduced with SynCAM 1-HRP rAAV and treated with biotin-AEEA-phenol omitting H_2_O_2_ (condition 4) or with H_2_O_2_ (condition 5).

For biochemical analysis, protein samples from treated neurons were separated and biotinylated proteins were visualized by Western blotting (Figure 3C). This showed that proteins were biotinylated in SynCAM 1-HRP or Membrane-HRP expressing neurons in presence, but not in absence, of H_2_O_2_ and conjugates spanned a large molecular weight range (Figure 3C, lanes 3, 5). Endogenously biotinylated proteins (e.g., histones and mitochondrial carboxylases) [29,59,60,61,62] were detected independent of the biotinylation reaction, as expected (Figure 3C). 

For proteomic analysis, four independent biological replicate experiments were performed and six pair-wise comparisons of protein levels of identified proteins across all four biological replicate experiments examined the robustness of this approach. As reliable measure for protein levels, intensity-based absolute quantification (iBAQ) levels were calculated [45] as iBAQ levels are proportional to molar abundance [46]. Pair-wise comparisons across biological replicates of protein iBAQ values or molar abundance of condition 5 showed strong correlation between biological replicates over minimally 5 orders of magnitude (Figure 3D). Several known synaptic proteins expected to be present at the synaptic cleft, including Neurexin-1, Neuroligin-3, Latrophilin-3, Contactin-1, Kilon, Hapln1, and Noelin-1, were found across the molar abundance spectrum. Retention of proteins due to interaction with other specifically bound proteins was unlikely as proteins were denatured before bead-binding. Further, endogenously biotinylated mitochondrial carboxylases and nuclear histones were among the most abundant proteins detected. The raw data also included intracellular proteins, e.g., GAPDH and Erlin-2, which likely unspecifically adsorbed to the beads. These non-synaptic hits identified in SynCAM 1-HRP samples further highlighted the need for a data filtering approach that analyzes a protein’s extent of biotinylation by a cleft reporter relative to a control to determine actual synaptic cleft abundance.

### 3.5. Ratiometric Analysis of Proximity Labeled Protein Hits

In silico post hoc filtering was implemented to identify biotinylated, synaptic cleft-enriched proteins, following a previously-described ratiometric approach [29]. Specifically, proteins identified in any of the conditions were sorted into four groups: (i) true positives, i.e., known synaptic proteins, TP1; (ii) false positives, i.e., known intracellular proteins, FP1; (iii) false positives, i.e., known surface proteins that are not synapse-enriched, FP2; (iv) all other proteins (Figure 4A). In filter step 1, the log_2_ ratio of normalized iBAQ values (i.e., relative to total iBAQ per sample; riBAQ) for condition 5 (c5) over condition 1 (c1) assessed how likely a protein was to be biotinylated by SynCAM 1-HRP. In filter step 2, the ratios of riBAQ for condition 5 (c5) over condition 3 (c3) was calculated to indicate how likely a protein was biotinylated by SynCAM 1-HRP compared to Membrane-HRP. As first internal quality control of the data filtering, the set of known synaptic surface proteins TP1 was used [29], for which c5/c1 riBAQ ratios are expected to be higher than for known intracellular proteins (FP1). As second internal quality control, the c5/c3 riBAQ ratios were assessed, which are expected to be higher for the list of synaptic membrane proteins (TP1) over the set of known surface proteins (FP2). Indeed, the distribution of c5/c1 riBAQ values of known synaptic proteins was bimodal with a first peak at the distribution of known intracellular proteins and a second peak at higher c5/c1 values (Figure 4B). Similarly, the distribution of known synaptic proteins was shifted towards higher c5/c3 riBAQ values compared with known surface proteins. As expected, the ratio of SynCAM 1-HRP plus H_2_O_2_ over SynCAM 1-HRP minus H_2_O_2_, i.e., c5/c4 riBAQ values, had comparable distributions as c5/c1 riBAQ values (not shown).

This analysis was expanded to select a cutoff for each filter step and obtain from the filtered data an enriched selection of proteins with maximal synaptic cleft proteins and minimal intracellular or dendritic surface proteins. Specifically, a Receiver Operating Characteristic (ROC) curve analysis was performed [47] to determine the optimal c5/c1 riBAQ and c5/c3 riBAQ cut-off values (Figure 4C). In brief, proteins in a biological replicate were ranked according to c5/c1 riBAQ values (filter 1) or c5/c3 riBAQ values (filter 2) (see Methods for ranking rules). For each protein, the rate of finding a false positive (False Positive Rate, FPR1 for intracellular protein and FPR2 for surface proteins) was subtracted from the rate of finding a true positive (True Positive Rate, TPR). TPR-FPR was plotted against the riBAQ values for each ranked protein (see Methods). At the maximum TPR-FPR, the matching ranked protein and corresponding c5/c1 riBAQ or c5/c3 riBAQ value was determined and used as minimum cut-off to retain proteins for filter 1 or 2, respectively (Figure 4C). 

In all four biological replicate experiments, a total of 706 proteins were identified (Figure 4D). After filter 1, total protein number was reduced to 50% (or 353 proteins), the group of known intracellular proteins was reduced by 50%, and the group of known surface proteins not enriched at synapses was reduced to 55%. All known synaptic proteins were retained. After filter 2, total protein number was reduced to 28% (or 200 proteins), known intracellular proteins to 27%, and known surface proteins to 20%, while 97% of known synaptic proteins were retained (Figure 4E). A fraction of proteins that passed filter 1 and 2 (33.5%) showed the same selection in multiple rounds of biological replicate experiments. A third filtering step was introduced that used the information for how frequently a protein passed filter 1 and 2 across biological replicates. Proteins were ranked according to c5/c3 riBAQ values and plotted against TPR-FPR2. The proteins that passed filter 1 and 2 in at least two of the four biological replicates showed a strong increase in TPR-FPR2, indicating that there was a strong enrichment for true positives (Figure 4F). 

Filter 3 encompassed this strict inclusion criterion to include only proteins that passed Filter 1 and 2 in minimally 2 biological replicates, and when applied, 67 proteins were retained (Figure 4E). After applying Filter 1 (selecting for biotinylated proteins), Filter 2 (selecting for synaptic cleft-biotinylated proteins), and Filter 3, 9% of total protein, 4% of known intracellular proteins, and 5% of known surface proteins were retained. Importantly, 70% of known synaptic proteins were still retained after all filters had been applied. It was expected that the resulting list still contained false positives, specifically intracellular proteins due to the designed filtering steps that minimized but not necessarily eliminated false positives. In a final selection step, proteins were removed that had only GO-terms associated with intracellular locations and proteins on the False Positive 1 (FP1)-list [29]. The resulting list contained 39 synaptic cleft candidate proteins (Appendix A). Twenty-six of the 123 proteins that were identified in 1 biological replicate neither had a GO-terms associated with intracellular locations nor were on the False Positive 1-list and were added to the list synaptic cleft-enriched candidates (Appendix A). Together, this resulted in a set of proteins with high confidence of synaptic cleft localization that were cleft-enriched in multiple biological replicates and a set of proteins that were enriched in one biological replicate.

### 3.6. Molecular Class and Gene Ontology Analysis of Synaptic Cleft Hits

The list of 39 proteins (Figure 5: black, underlined gene names) found in multiple biological replicates and the 26 additional proteins (Figure 5, grey gene names) found in one biological replicate were characterized for Molecular Class according to the Human Protein Reference Database (HPRD) [63,64,65]. The HPRD classifies each protein into one category, which simplifies functional protein characterization. Proteins were classified across major categories expected in the synaptic cleft: 24% of proteins were associated with either adhesion molecule (18%), immunoglobulin (3%) or adhesion molecule activity (3%); 20% of proteins were associated with cell surface receptor (14%), G protein-coupled receptor (3%), or receptor tyrosine phosphatase (3%); 20% of proteins were associated with either Membrane transport protein (11%), Extracellular ligand-gated channel (5%), Ion channel (2%), or Voltage-gated channel (2%); 14% were associated with membrane transport protein (11%) or Transport/cargo protein (3%). 

Previous proteomic studies of synapses have targeted different synaptic compartments or protein complexes [11,15,16,17,18,19,20,21,22,66,67,68]. A comparison with these studies found that almost all of 65 cleft-enriched proteins identified here, except for Contactin-5, were reported earlier to be synaptic. However, none of these studies specifically characterized the synaptic cleft. The only study so far targeting the synaptic cleft by Loh and colleagues had an estimated 69% excitatory synaptic cleft proteome coverage and reported several novel synaptic cleft candidates [29]. Compared with this previous work, 30 proteins were solely enriched in this study using SynCAM 1-HRP-mediated proximity labeling of the synaptic cleft (Appendix A). Eleven synaptic cleft candidates identified here were neither enriched in the previous proteomic study targeting the (excitatory and inhibitory) synaptic cleft [29], nor did they contain a GO-term associated with synapses. Several of these proteins, such as CD166 antigen and Leucine-rich glioma-inactivated protein 1, were identified in previous proteomic studies of the active zone and other synaptic compartments [11,15,16,17,18,19,20,21,22,66,67,68]. However, these studies did not demonstrate that these 17 proteins were synaptic cleft-enriched, and a literature and GO-term analysis did not find that these were synaptic cleft-enriched proteins. Hence, these proteins were termed candidate excitatory synaptic cleft orphans (Appendix A). 

### 3.7. Validation of the Synaptic Cleft Candidate Receptor-Type Tyrosine-Protein Phosphatase Zeta

To validate the peroxidase-mediated proximity labeling approach and filtering process for identification of novel synaptic cleft proteins, a candidate protein not previously identified in a synaptic cleft proteomics study (Appendix A) was selected for further study. Receptor-type tyrosine-protein phosphatase zeta, or R-PTP-zeta (gene name: *Ptprz1*) was previously detected in proteomic studies of postsynaptic density and synaptosomes of rodent and human brain [15,20,21,22]. Here, coronal sections of mouse brain stained for R-PTP-zeta showed immunofluorescence in hippocampus and cortex (Figure 6). Immunostaining for R-PTP-zeta was particularly strong in areas with strong immunostaining for postsynaptic marker Homer and presynaptic marker Bassoon, suggesting that R-PTP-zeta is associated with excitatory synapses.

Indeed, immunostaining of cultured rat cortical neurons for surface-expressed R-PTP-zeta under non-permeabilized conditions showed its punctate localization along dendrites. R-PTP-zeta puncta colocalized with puncta positive for the presynaptic marker Bassoon. Moreover, these puncta were also immunopositive for surface-expressed SynCAM 1 (Figure 7A), as expected for a protein identified by proximity-reporter SynCAM 1-HRP. Similarly, extracellular R-PTP-zeta colocalized with puncta positive for the excitatory postsynaptic marker Homer that were also positive for extracellular SynCAM 1 (Figure 7B). These results agree with a previous study that used antibodies recognizing R-PTP-zeta to detect immunoreactivity at PSD-95-positive spines of pyramidal neurons in cerebral cortex and hippocampus of rats, specifically the postsynaptic membrane of dendritic spines and shafts [69]. Our results support that endogenous SynCAM 1 and R-PTP-zeta are co-expressed at the cleft of the same excitatory synapses and may be in close proximity, in agreement with the proteomic identification of R-PTP-zeta by SynCAM 1-HRP.

## 4. Discussion

Brain circuits are anatomically and functionally highly diverse, which corresponds with the different neuronal cell types found across regions [70]. In agreement with the distinct expression profiles of these neuron types, the synapses they form are highly heterogenous in function and composition [2]. Specifically, the postsynaptic proteome of the brain exhibits unique compositional signatures, which correlate with anatomical divisions of the brain both in mice [71,72] and humans [73,74]. Within a brain region, inputs originating from different neuronal populations may synapse onto one particular neuron and these inputs on the same target have specific functional characteristics [75,76,77,78]. This involves synapse-organizing mechanisms to which synaptic cell-surface proteins such as adhesion proteins of the immunoglobulin and leucine-rich repeat protein super-families contribute in the hippocampus [79]. Instructive roles of adhesion molecules in synapse specification are underlined by the roles of immunoglobulin proteins and cadherins in shaping connectivity in the retina [80,81]. Hence, synaptic cleft proteins are positioned to play an essential role in establishing synapse connectivity and function and specify the identity of synapses. It is therefore important to determine to what extent synapses differ based on the cleft proteins they contain. Moreover, the ability to answer this question allows to assess how disease states or substance abuse remodel the cleft and change its synapse-organizing properties. 

This study applied a peroxidase-mediated proximity labeling approach to examine the protein composition of excitatory synaptic clefts in cultured cortical neuron [25]. The excitatory synaptic cell adhesion protein SynCAM 1 of the immunoglobulin superfamily was utilized as a new reporter to target HRP to synaptic clefts of glutamatergic synapses and label and identify its proteomic content. This generated a list of synaptic cleft candidates of which several are novel. The proximity labeling approach is robust as shown by its reproducibility across biological replicates and stringent in silico data filtering. The list of proteins this approach identified in our study was enriched for synaptic membrane proteins and depleted for general surface proteins and intracellular contaminants. 39 proteins were detected as synaptic cleft candidates in multiple biological replicates, and 26 additional proteins were found in only one of the four biological replicates. Of these 65 proteins, 30 proteins are novel compared with a previous study that used this approach to map the proteome of excitatory synaptic clefts with different HRP reporters [29] (Appendix A). These novel cleft candidates included adhesion proteins, receptors, and secreted proteins/extracellular matrix proteins. The fact that a differential set of proteins was enriched, despite this study having a lower estimated coverage (16%) of the excitatory cleft proteome compared with Loh et al. may be explained by several factors. First, SynCAM 1 (this study) and LRRTM1/2 (used to design reporters by Loh et al.), may have differential expression across synapse types. SynCAM 1 is expressed in the cortex of rodent brain [35,36], similarly, LRRTM1-2 are expressed in the cortex [82,83]. Yet, it is unclear to what extent these proteins may be expressed at the same synapse. Second, the possibility exists that SynCAM 1 and LRRTMs localize to different sub-cleft regions and that reporters based on these molecules probe different sub-synaptic cleft proteomes. Endogenous SynCAM 1 was primarily found at the edges of the area marked by the postsynaptic density [14] consistent with the results obtained in this study for SynCAM 1-HRP, while LRRTM2 resides more closer towards the center of the postsynaptic density [84]. The radius of biotinylated proteins proximal to peroxidase-fusion proteins after proximity labeling is as previously reported to be <20 nm [28,55], which is a fraction of the synaptic cleft length [12,85,86]. Hence, HRP-fusion proteins of these reporters may probe differential environments within the synaptic cleft, opening the possibility to map sub-cleft proteomes and increase the molecular definition of the cleft.

The identification of synaptic cleft candidates that were previously not described calls for their validation to attest to proximity labeling as a tool to map cleft proteomes. The protein selected in this study for validation, Receptor-type tyrosine-protein phosphatase zeta, or R-PTP-zeta (gene product of *Ptprz1*) was enriched in three out of four biological replicate experiments providing confidence in a possible synaptic cleft localization. R-PTP-zeta is predominantly expressed in the central nervous system and plays a role during development and adulthood in myelination and learning and memory processes [87,88], and *PTPRZ1* may be a potential schizophrenia susceptibility gene [89,90,91]. R-PTP-zeta (RPTPζ/β) was previously detected at some PSD-95-positive spines of pyramidal neurons in cerebral cortex and hippocampus of rats, specifically at the postsynaptic membrane of dendritic spines and at shafts [69]. R-PTP-zeta is expressed as three structurally distinct isoforms: a long, membrane-integral isoform; a short, membrane-integral isoform; and a soluble isoform [92,93,94]. While it is unclear which specific variant was found enriched in this proteomics study, in situ hybridizations of Ptprz1 mRNA by the Allen Institute for Brain Science suggest that the long isoform of R-PTP-zeta is mainly expressed in the olfactory areas and the cerebellum. A closer examination finds that these probes match a region of exon 12 that is excluded from the short membrane isoform. Results here show that R-PTP-zeta is expressed in cortical neurons in dissociated cultures derived from cortices and cortical and hippocampal regions of the mouse brain (Figure 6 and Figure 7). Hence, the short isoform is likely the variant detected by our immunohistochemical staining in the cortex and hippocampus. Notably, we observed in dissociated cortical neurons a strong co-localization of R-PTP-zeta with synaptic markers, including the excitatory marker Homer 1, and with endogenous SynCAM 1. These results validate the identification of R-PTP-zeta in this screen.

Together, our data support that proximity labeling using synaptic HRP reporters is a robust approach to identify the molecular composition of the cleft, a compartment not readily accessible to previous biochemical studies. Future applications can include testing changes in the makeup of the synaptic cleft under disease-linked conditions that alter synapse structure, e.g., mouse models relevant for developmental disorders. Moreover, synapse structure is altered by repeat administration of psycho-stimulants, and the synapse organizer SynCAM 1 acts in medium spiny neurons to control the number of their dendritic spines and the remodeling of spine morphology in these neurons upon cocaine exposure [95]. This provides evidence that cleft components can contribute to the synaptic changes upon exposure to drugs of abuse and warrants future proteomic studies to map these changes. Moreover, peroxidase-mediated proximity labeling offers the opportunity for performing acute manipulations to measure acute synaptic cleft remodeling. Recently, an improved BioID method has been introduced, TurboID, which has a temporal resolution that is in the order of tens of minutes and may be used to measure long-term remodeling of the synaptic cleft proteome in vivo. These approaches can therefore be utilized in future studies to analyze the activity-dependent re-organization of the synaptic cleft that is supported by the redistribution of SynCAM 1 and Neuroligin-1 after induction of long-term depression [14,84]. Proximity labeling of cleft proteins may hence provide valuable insights into roles of the dynamic cleft in shaping synapses [13].

## Figures and Tables

**Figure 1 proteomes-06-00048-f001:**
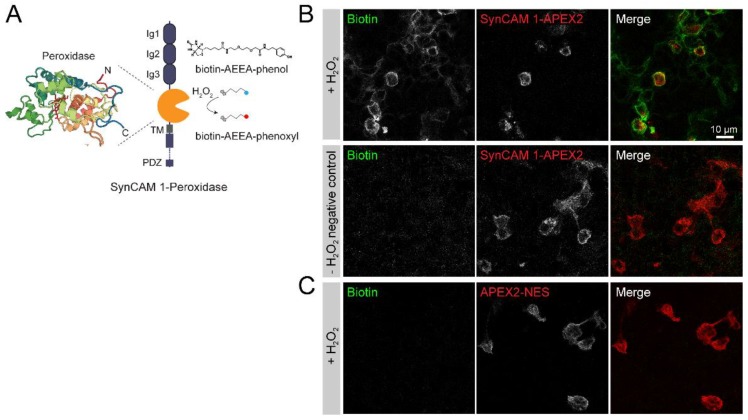
SynCAM 1-peroxidase fusion protein peroxidase-mediated proximity labeling in the synaptic cleft. (**A**) APEX2 or HRP (image RCSB PDB [53,54] (www.rcsb.org) of PDB ID 1HCH [54]) peroxidase was inserted at the base of the SynCAM 1 extracellular domain, with immunoglobulin (Ig) domains, trans-membrane (TM) region, and intracellular PDZ domain interaction sequence indicated. APEX2 or HRP catalyzes the formation of a short-lived biotin-AEEA-phenoxyl radical (red dot) after exogenous addition of H_2_O_2_ and membrane-impermeable biotin-AEEA-phenol (blue dot). (**B**) Exogenous biotin-AEEA-phenol induced biotinylation only at the cell surface. Staining for biotin (visualized by StreptAvidin-Alexa488) in HEK293T cells expressing SynCAM 1-APEX2 in presence (+) but not in absence (−) of H_2_O_2_. (**C**) Exogenous biotin-AEEA-phenol did not induce biotinylation in HEK293T cells expressing cytosolic APEX2-NES.

**Figure 2 proteomes-06-00048-f002:**
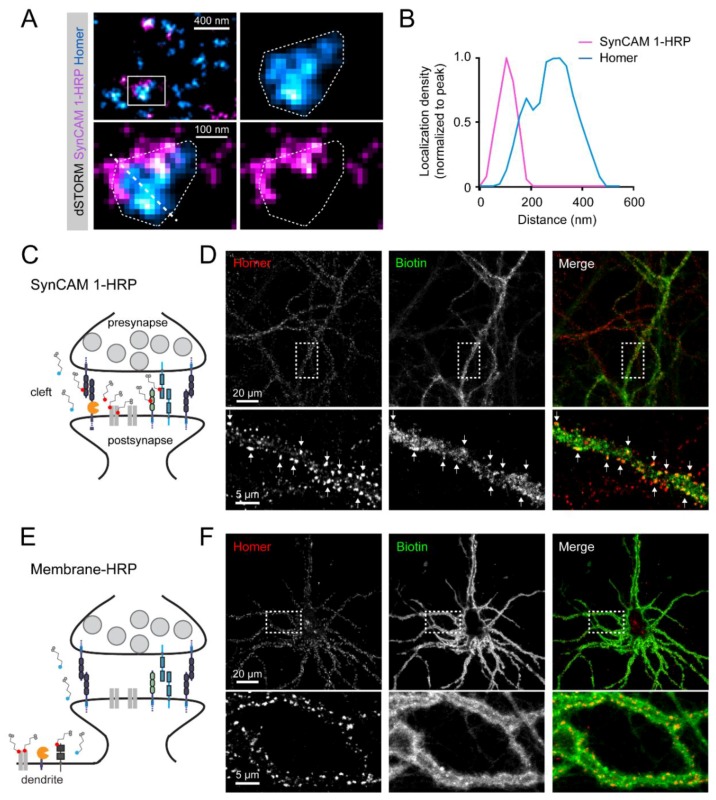
Synaptic SynCAM 1-HRP expression and biotinylation. (**A**) Two-color dSTORM reconstruction of synapses from 21 days-in-vitro (div) rat hippocampal neurons surface-labeled by immunostaining with anti-FLAG antibodies against exogenous SynCAM 1-HRP containing the FLAG epitope (magenta) and the endogenous excitatory postsynaptic marker Homer (cyan). Top left, overview. Enlarged panels show one synapse with SynCAM 1-HRP and Homer localizations fit by a convex hull to demarcate the PSD border (dotted outline). SynCAM 1-HRP localizations are at the periphery of the PSD. Diagonal line, line scan used for (**B**). See Appendix A for additional examples and quantitative analyses. (**B**) Protein localization distribution perpendicular to the trans-synaptic axis. Densities were determined by dSTORM and normalized to the peak of each channel and measured over the distance shown in A. (**C**) Model of HRP targeting by the reporter SynCAM 1-HRP to excitatory synaptic clefts for biotinylation of proximal surface proteins. (**D**) Following the proximity labeling reaction with membrane-impermeant biotin-AEEA-phenol, biotin staining was detected in SynCAM 1-HRP transduced rat cortical neurons along dendrites at excitatory synaptic sites visualized by immunostaining of Homer (arrows). (**E**) Targeting of HRP by the reporter Membrane-HRP to the plasma membrane of dendrites. (**F**) Staining for biotin was detected in Membrane-HRP transduced rat cortical neuronal cultures along dendrites after proximity labeling with biotin-AEEA-phenol.

**Figure 3 proteomes-06-00048-f003:**
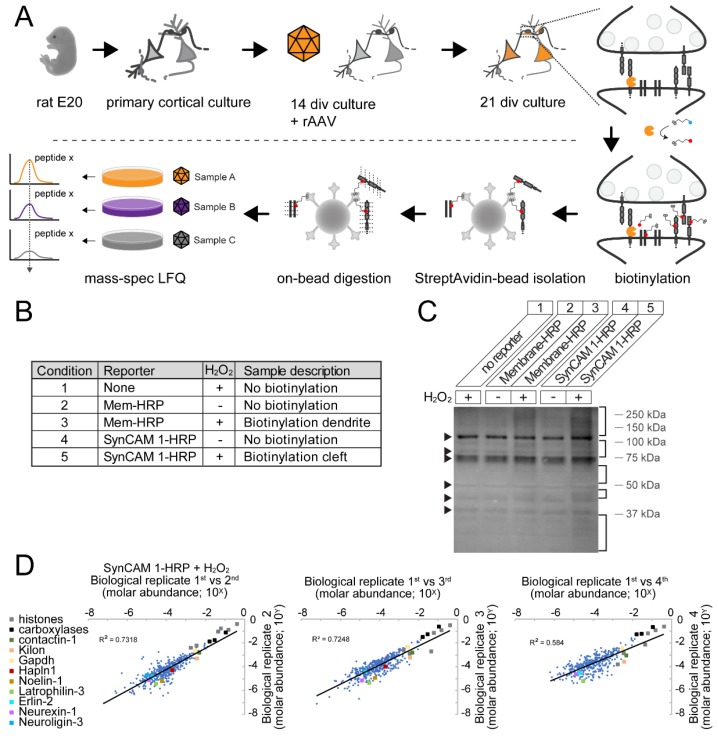
Scaled proximity labeling in cultured neurons. (**A**) Rat cortical neurons were transduced with rAAV encoding HRP reporters (orange) at 14 div and underwent proximity labeling at 21 div. SynCAM 1-HRP (orange) biotinylates synaptic proteins that are purified and digested for LFQ. (**B**) Each biological replicate included samples from the 5 conditions shown. (**C**) Western Blot of samples visualized by NeutrAvidin-DyLight488. Endogenously biotinylated proteins (arrow heads) are marked and weight ranges (brackets) wherein exogenously biotinylated proteins were detected are indicated. (**D**) Example comparisons of molar abundance ranges. Four biological replicates were performed and molar abundance (relative iBAQ or riBAQ) of all detected proteins in SynCAM 1-HRP samples were compared. Graphs show pair-wise comparisons of the 1st biological replicate with the 2nd, 3rd, and 4th replicate, respectively. Endogenously biotinylated proteins (histones, mitochondrial carboxylases), cytosolic proteins (GAPDH, Erlin-2) and synaptic cleft proteins (Contactin-1, Kilon, Hapln1, Noelin-1, Latrophilin-3, Neurexin-1, Neuroligin-1) are indicated.

**Figure 4 proteomes-06-00048-f004:**
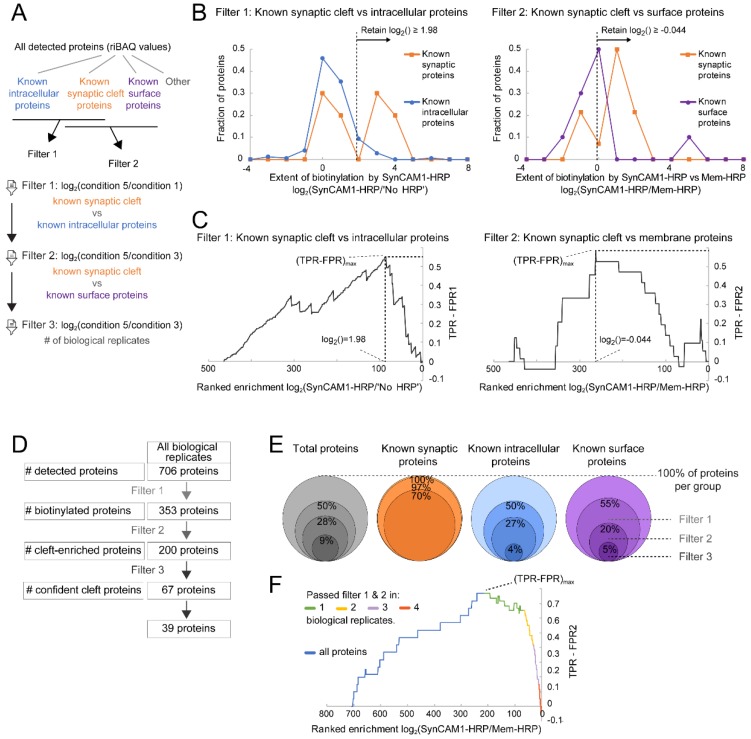
Data filtering based on proteins of known sub-cellular localizations. (**A**) Detected proteins were sorted into 4 categories based upon literature and these categories were used in Filter 1 and 2. Filter 3 used the number of biological replicates a protein passed Filter 1 and 2. (**B**) Histograms of the extent of biotinylation by SynCAM 1-HRP vs. no reporter control for Filter 1 (left plot) and vs. Mem-HRP for Filter 2 (right plot). Cut-off values determined in (**C**) for Filter 1 and Filter 2 are indicated. (**C**) Receiver Operating Characteristic (ROC) curve analysis to determine optimal enrichment for Filter 1 (left plot) or Filter 2 (right plot) by plotting the True Positive Rate (TPR: known synaptic proteins) minus the False Positive Rate (FPR). FPR1: known intracellular proteins; FPR2: known membrane proteins. At maximum (TPR-FPR), the log_2_ value is determined at the corresponding protein in the ranked protein list, which served as a cut-off value in (**B**). (**D**) Enrichment for excitatory synaptic cleft proteins through filtering. All proteins combined in four biological replicate experiments were subjected to Filter 1, Filter 2, and Filter 3. A final manual curation step removed remaining false positives. (**E**) Relative enrichment per group of proteins of known sub-cellular localization. Per group, all proteins (100% per group) are depicted as a circle of unitary size. Each filter step reduces protein number and circle area proportionally. Note that after Filter 1, the total proteins in group of known synaptic proteins is 100% and area of circle is unitary. (**F**) ROC analysis of TPR-FPR2 for Filter 3. Indicated are proteins that passed Filters 1 and 2 four times (red), three times (purple), two times (yellow), and one time (green).

**Figure 5 proteomes-06-00048-f005:**
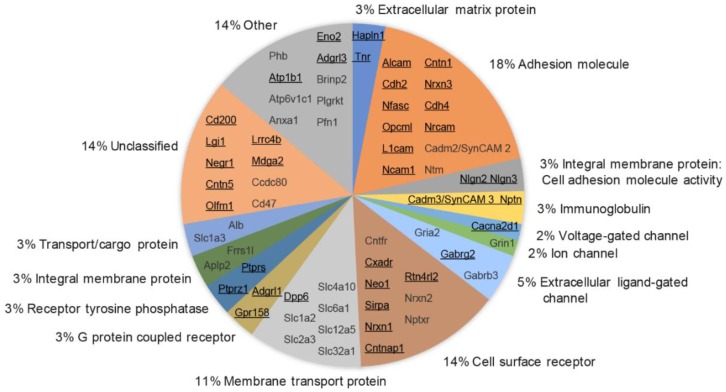
Pie chart showing relative abundance of protein functional groups (categories according to the Human Protein Reference Database) and synaptic cleft candidates in each group. Synaptic cleft candidates found in minimally two biological replicates are underlined. Candidates found only in one biological replicate are shown in grey font.

**Figure 6 proteomes-06-00048-f006:**
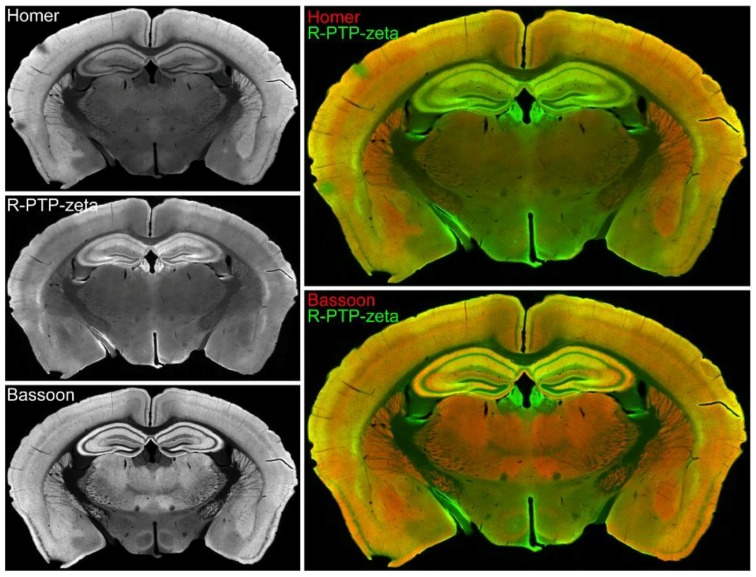
Characterization of R-PTP-zeta expression in vivo. Left, Immuno-histochemical staining of rat brain at P66 for the presynaptic marker Bassoon, excitatory postsynaptic marker Homer and R-PTP-zeta (grey in single color images). Right, yellow in the merged composite (top) image indicates co-expression of Homer (red) and R-PTP-zeta (green). Yellow in the bottom merged image indicates co-expression of Bassoon (red) and R-PTP-zeta (green).

**Figure 7 proteomes-06-00048-f007:**
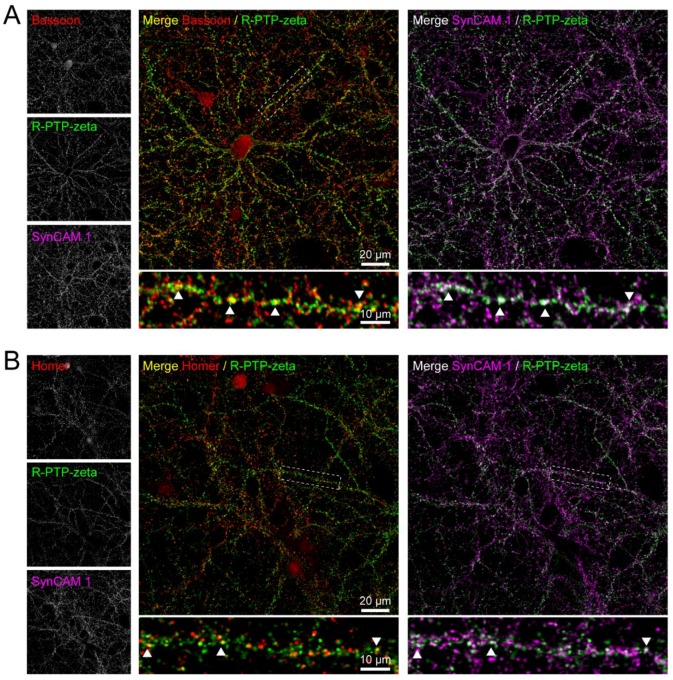
Characterization of R-PTP-zeta expression and synaptic markers in vitro. (**A**) Immunostaining for Bassoon, R-PTP-zeta, and SynCAM 1 in cultured neurons. R-PTP-zeta (grey in single color image) and SynCAM 1 (grey in single color image) were stained under non-permeabilizing conditions using antibodies detecting extracellular epitopes and Bassoon (grey in single color image) was stained under permeabilizing conditions. Yellow in larger composite (left) image indicates colocalization of Bassoon (red) and extracellular R-PTP-zeta (green). White in larger composite (right) image indicates colocalization of extracellular SynCAM 1 (magenta) and extracellular R-PTP-zeta (green). Panels below show enlarged dendritic segments from the composite images to visualize colocalization (arrowheads). (**B**) Immunostaining for Homer and R-PTP-zeta and SynCAM 1 in cultured neurons. Extracellular R-PTP-zeta (grey in single color image) and SynCAM 1 (grey in single color image) were immunostained under non-permeabilizing conditions as in (**A**) and Homer (grey in single color image) was immunodetected under permeabilizing conditions. Yellow in larger composite image (left) indicates colocalization of Homer (red) and extracellular R-PTP-zeta (green). White in larger composite image (right) indicates colocalization of extracellular SynCAM 1 (magenta) and extracellular R-PTP-zeta (green). Panels below show enlarged dendrites from the composite images. Arrowheads mark sites of colocalization.

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
