# Peer review of "Mapping the Proteome of the Synaptic Cleft through Proximity Labeling Reveals New Cleft Proteins"

_proteomes, 2018, doi:10.3390/proteomes6040048_

Round 1

Reviewer 1 Report

This paper describes the application of peroxidase-mediated proximity labelling to study the excitatory synaptic cleft proteome in neuronal cell culture. This involved the use of a SynCAM 1 fused to horseradish peroxidase (SynCAM1-HRP). A membrane-impermeant biotin-phenol compound was applied to induce HRP-catalyzed biotinylation of SynCAM1-HRP neighbouring proteins. Biotinylated proteins were then enriched with streptavidin beads and identified by label-free quantitation (LFQ) mass spectrometry. In silicofiltering steps were used to compare SynCAM1-HRP and membrane-HRP proteomes. Using this method the authors identified 65 proteins as synaptic cleft candidates. This is not a novel technique, it was first used for proximity labelling of excitatory synapse specific LRRTM1 and LRRTM2, and inhibitory synapse specific Slitrk3 and Nlgn2 in 2016 (Loh et al. 2016). This new work builds upon the published results by applying this method to another excitatory synaptic cleft specific protein. 30 of the synaptic cleft candidates identified here were novel and not identified in the original 2016 paper. Overall this work is a useful addition to the existing literature on the synaptic cleft proteome. It demonstrates the application of peroxidase-mediated proximity labelling to the study of this synaptic sub-region by an independent group of researchers, thus validating its use as a tool for this kind of research. The paper is generally well written, methods are detailed, and conceptual figures are used to good effect to explain experimental design. The authors validated their experiment by investigating one novel synaptic cleft candidate protein in more detail.

Points to address

1.    It would be good if the basic concept of peroxidase-mediated proximity labelling could be introduced earlier than it is, perhaps around line 60. A simple schematic like that in the lou et al. 2016 paper showing the localisation of peroxidase-generated biotin labels to the synaptic cleft would make this clearer from the start. 

2.    There does not appear to be any discussion of the expected radius of proximity labelling using this method. This would be useful information to have. 

3.    Is the addition of SynCAM1-HRP likely to have any effect on synapse structure? It might be useful to have an experiment to compare pre- and post-synaptic marker distribution in control and SynCAM1-HRP positive cultures. 

4.    The data from the dSTORM imaging presented in figure 2A & 2B would be more convincing if more than one representative synapse was shown. Some quantification of the number of SynCAM1 puncta juxtaposed with homer would be beneficial. 

5.    Figure 2D shows biotin staining at confocal resolution but there appears to be a significant portion of biotin staining signal that does not colocalise with homer. This figure would benefit from some quantification of puncta colocalisation and some discussion of this homer independent staining in the text would be welcome. 

6.    Similarly, the validation of R-PTP-zeta as a synaptic cleft protein is essential to the confirmation of the ability of this experiment to identify novel synaptic cleft proteins. The current immunochemistry data would be significantly more convincing with the addition of some quantification of puncta colocalisation, to establish the proportion of R-PTP-zeta immunofluorescent puncta that are colocalised with synaptic markers. The gold standard here would obviously be immuno-EM to identify the precise location of R-PTP-zeta. 

7.    It is not clear in figure 3C that the intense bands visible across every lane are endogenously biotinylated proteins. The addition of some arrows to that figure and some explanation in the figure legend would make this easier to understand. A reference which explains what these bands are likely to consist of would also be useful.   

8.    A diagram of the filter steps as described in lines 581-585 would be useful. Figure 4A goes some way towards this but it would be clearer if the conditions involved in each step could be highlighted. E.g. membrane-HRP biotinylation/SynCAM1-HRP biotinylation etc. 

9.    Could the authors elaborate on the comment in line 760-761 on potential differential expression of SynCAM1 and LRRTM? Are there any references to suggest this? As far as I can tell both studies used dissociated rat neuronal cultures at the same timepoint so I’m unsure why a developmental change in either protein would affect the results. 

Author Response

Response to Reviewer 1 Comments

We thank the Reviewer for the comprehensive review of our manuscript and the thoughtful comments she/he has provided. We are pleased the reviewer can conclude that our presented results are “a useful addition to the existing literature on the synaptic cleft” and finds that our work validates the method of peroxidase-mediated proximity labeling as a tool to map the proteome of the synaptic cleft.  We also appreciate that the Reviewer acknowledges our efforts to present the methodology in detail. We have performed the requested additional image analyses and provide them in the new Supplemental Figure S1 and addressed the Reviewer’s other comments below in blue font. We hope she/he can conclude that these revisions significantly strengthen this study.

Points to address

1.    It would be good if the basic concept of peroxidase-mediated proximity labelling could be introduced earlier than it is, perhaps around line 60. A simple schematic like that in the lou et al. 2016 paper showing the localisation of peroxidase-generated biotin labels to the synaptic cleft would make this clearer from the start. 

Response 1: We appreciate the Reviewer’s suggestion that the concept of peroxidase-mediated proximity labeling can be introduced earlier in the manuscript. A graphical abstract that accompanies the manuscript has now been added to introduce the approach at the beginning of the manuscript.

2.    There does not appear to be any discussion of the expected radius of proximity labelling using this method. This would be useful information to have.

Response 2: We have added information on the expected radius of proximity labeling using the SynCAM 1-HRP reporter to the Discussion (Line 777 of the revised manuscript). The radius of ~20 nm is expected to be the same as previously described (Loh et al 2016) as it is mostly dependent on the life-time of biotin radicals.

3.    Is the addition of SynCAM1-HRP likely to have any effect on synapse structure? It might be useful to have an experiment to compare pre- and post-synaptic marker distribution in control and SynCAM1-HRP positive cultures. 

Response 3: We acknowledge the Reviewer’s concern of effects on synapse structure by exogenous expression of SynCAM 1-HRP and have analyzed Homer puncta area in SynCAM 1-HRP transfected neurons using dSTORM experiments. This did not determine obvious structural aberrations in Homer puncta compared to other studies in the field and in the Blanpied lab, see Supplemental Figure S1. Our approach to analyze Homer puncta area is added to the Supplementary Materials.

4.    The data from the dSTORM imaging presented in figure 2A & 2B would be more convincing if more than one representative synapse was shown. Some quantification of the number of SynCAM1 puncta juxtaposed with homer would be beneficial. 

Response 4: We show in the new Supplementary Figure S1 additional representative synapses with SynCAM 1-HRP clusters juxtaposed to Homer clusters. In addition, we now include in this figure a new quantification of SynCAM 1-HRP cluster distance to Homer cluster center. These SynCAM 1-HRP to Homer distances are in agreement with previous results of the distance of endogenous SynCAM 1 to Homer (de Arce et al, 2015). To address the second sub-point, we have quantified the number of SynCAM 1 clusters juxtaposed with Homer and added that analysis to Supplementary Figure S1.

5.    Figure 2D shows biotin staining at confocal resolution but there appears to be a significant portion of biotin staining signal that does not colocalise with homer. This figure would benefit from some quantification of puncta colocalisation and some discussion of this homer independent staining in the text would be welcome. 

Response 5: We agree with the Reviewer’s observation that biotin staining is detected in SynCAM 1-HRP expressing neurons at membrane sites lacking Homer signal. This may be caused by SynCAM 1-HRP being partially localized at extra-synaptic sites. We now point this out more clearly in the revised text (Line 512). We have considered a colocalization analysis of biotin signal generated by SynCAM 1-HRP with synaptic markers to evaluate the extent of overlap with Homer. While we clearly observe colocalization, we decided not to include a quantification in the manuscript due to the low signal-to-noise ratio in these fluorescence images; we would have had to rely on rather difficult to determine cut-offs to differentiate signals that are synaptic or not. In response to the reviewer’s point, we have added that partial biotin-labeling is observed (and perhaps even can be expected) outside the excitatory synaptic cleft and that this further warrants the need for the implemented ratiometric comparison of synaptic cleft-labeled proteins with membrane-labeled proteins (Section 3.5).

6.    Similarly, the validation of R-PTP-zeta as a synaptic cleft protein is essential to the confirmation of the ability of this experiment to identify novel synaptic cleft proteins. The current immunochemistry data would be significantly more convincing with the addition of some quantification of puncta colocalisation, to establish the proportion of R-PTP-zeta immunofluorescent puncta that are colocalised with synaptic markers. The gold standard here would obviously be immuno-EM to identify the precise location of R-PTP-zeta.

Response 6: The Reviewer is correct to note that immuno-EM is the gold standard to identify the precise localization of synaptic proteins. We have analyzed our immunostaining data and find that approximately half of R-PTP-zeta puncta colocalize with endogenous SynCAM 1 and similarly half of R-PTP-zeta puncta colocalize with synaptic markers. These findings are in line with the expectations based on the identification of R-PTP-zeta using synaptic cleft proximity labeling. This analysis was only from a single round of experiments, though, and we have therefore not added this data to the manuscript. Future studies can now focus on the precise localization of R-PTP-zeta, particularly with respect to the synaptic localization of SynCAM 1.

7.    It is not clear in figure 3C that the intense bands visible across every lane are endogenously biotinylated proteins. The addition of some arrows to that figure and some explanation in the figure legend would make this easier to understand. A reference which explains what these bands are likely to consist of would also be useful.   

Response 7: The labeling of Figure 3C, including its legend, has been revised as suggested to indicate bands that we interpret as endogenously biotinylated proteins based on their presence in control conditions (minus H2O2). Endogenous mitochondrial carboxylases and nuclear histones can be biotinylated and are likely the bulk of proteins identified in this control condition and we now cite relevant references (Line 562). We have also added brackets to indicate the molecular weight range of exogenously biotinylated proteins detected upon proximity labeling. 

8.    A diagram of the filter steps as described in lines 581-585 would be useful. Figure 4A goes some way towards this but it would be clearer if the conditions involved in each step could be highlighted. E.g. membrane-HRP biotinylation/SynCAM1-HRP biotinylation etc. 

Response 8: This suggestion is very helpful to explain the approach, and a diagram has been added to Figure 4A to clarify the filtering steps and the conditions involved in each step.

9.    Could the authors elaborate on the comment in line 760-761 on potential differential expression of SynCAM1 and LRRTM? Are there any references to suggest this? As far as I can tell both studies used dissociated rat neuronal cultures at the same timepoint so I’m unsure why a developmental change in either protein would affect the results. 

Response 9: We agree with the Reviewer that there are insufficient grounds to assert that differential developmental expression could explain these results, and we have removed this statement from the Discussion (Line 773).

Additional revisions:

1.      A graphical/visual abstract has been added to accompany the revised manuscript.

2.      Section 2.4 Plasmids contained an erroneous plasmid including its cloning description. This has been replaced with plasmid pCAGGS-SynCAM 1-APEX2 and proper cloning description. Other updates in this section are reflected as well, including Addgene plasmid IDs.

3.      In Figure 1 legend, citations have been added for the use of the HRP protein model from TCSB PDB.

4.      In Figure 4B, a description has been updated.

5.      From Table S3, Serum albumin was removed as it is likely a carry-over from cell culture media. Text referencing the number of candidates in Table S3 has been updated.

6.      The colors in Figure 5 were adjusted for better readability of the text.

7.      A methods section was added in the main text accompanying the new analyses now shown in Supplemental Figure S1.

Reviewer 2 Report

The article by Cijsouw and Biederer and colleagues represents a novel proximity labeling data set analyzing the synaptic cleft proteome. This data set shows multiple levels of filtering to attain a specific synaptic cleft proteome isolated from rat cultures. The data are well controlled and the reagents well validated and they add new information to the synaptic cleft proteome. Overall, the manuscript is well-written and understandable. However, there are a couple of minor issues that I would like some clarification if possible.

1.     In section 2.9.3, the authors mention c5/c1 and c5/c3. While they define these later (in figure 3B and in the results, it would be nice to define the conditions here. This could be done by referencing figure 3B or moving figure 3B to a new table in the methods (and keeping or eliminating it from the figure based on the author’s likes). It was a little confusing reading about this here without the background information supplied in the results.

2.     In section 3.3, the authors say “rat cortical neurons…” but in the figure legend for Figure 2 they mention rat hippocampal neurons, at least for panel A. Were panels A hippocampal and D and F cortical neurons? If so, maybe define which panel was which in both the results and legend.

3.     In the discussion, the authors mention the challenge of acute manipulations using BioID. They may want to reference newer, recently developed/developing approaches using TurboID and miniTurbo that use more robust and faster (e.g. minutes) forms of the BioID molecule (Branon et al., Efficient proximity labeling in living cells and organisms with TurboID. Nature Biotechnology 2018) and how these may be useful for the types of approaches outlined in the current manuscript.

Author Response

Response to Reviewer 2 Comments

We thank the Reviewer for the thorough reading of our manuscript and the valid points raised. We are pleased that she/he concludes “the data are well controlled and reagents well validated” and that this study adds “new information to the synaptic cleft proteome”. We have addressed the points raised by the Reviewer below in blue font and hope she/he can conclude that these revisions further strengthen the manuscript.

Points to address

1.     In section 2.9.3, the authors mention c5/c1 and c5/c3. While they define these later (in figure 3B and in the results, it would be nice to define the conditions here. This could be done by referencing figure 3B or moving figure 3B to a new table in the methods (and keeping or eliminating it from the figure based on the author’s likes). It was a little confusing reading about this here without the background information supplied in the results.

Response 1: We agree with the Reviewer that this information is essential for understanding the method and the conditions need to be defined at the first mentioning of the conditions. The definitions of experimental conditions were added to section 2.9.3 (Line 277 of the revised manuscript) to address this issue.

2.     In section 3.3, the authors say “rat cortical neurons…” but in the figure legend for Figure 2 they mention rat hippocampal neurons, at least for panel A. Were panels A hippocampal and D and F cortical neurons? If so, maybe define which panel was which in both the results and legend.

Response 2: As the Reviewer was correct to note, Figure 2A-B were performed in rat hippocampal neurons (to compare with previous results in Arce et al) and Figure 2C-F were performed in rat cortical neurons. This has now been clarified in both the results (section 3.2) and the legend (Figure 2).

3.     In the discussion, the authors mention the challenge of acute manipulations using BioID. They may want to reference newer, recently developed/developing approaches using TurboID and miniTurbo that use more robust and faster (e.g. minutes) forms of the BioID molecule (Branon et al., Efficient proximity labeling in living cells and organisms with TurboID. Nature Biotechnology 2018) and how these may be useful for the types of approaches outlined in the current manuscript.

Response 3: We thank the Reviewer to pointing us to this excellent paper by Branon et al. on the recently developed TurboID. Indeed, TurboID has much faster labeling compared with BioID. TurboID is now introduced in the Introduction (Line 64). The use of TurboID is then discussed in the Discussion (Line 812) of the revised manuscript) as an alternative approach to measure long-term synaptic cleft proteome remodeling in vivo or in vitro.

Additional revisions:

1.      A graphical/visual abstract has been added to accompany the revised manuscript.

2.      Section 2.4 Plasmids contained an erroneous plasmid including its cloning description. This has been replaced with plasmid pCAGGS-SynCAM 1-APEX2 and proper cloning description. Other updates in this section are reflected as well, including Addgene plasmid IDs.

3.      In Figure 1 legend, citations have been added for the use of the HRP protein model from TCSB PDB.

4.      In Figure 4B, a description has been updated.

5.      From Table S3, Serum albumin was removed as it is likely a carry-over from cell culture media. Text referencing the number of candidates in Table S3 has been updated.

6.      The colors in Figure 5 were adjusted for better readability of the text.

7.      A methods section was added in the main text accompanying the new analyses now shown in Supplemental Figure S1.